# Ecological Urban Planning and Design: A Systematic Literature Review

**Angela Heymans, Jessica Breadsell** * **, Gregory M. Morrison, Joshua J. Byrne and Christine Eon**

Curtin University Sustainability Policy Institute, School of Design and the Built Environment, Curtin University, Bentley 6102, Australia

*   Correspondence: Jessica.breadsell@curtin.edu.au

**Abstract:** Urbanization is a defining feature of the modern age, yet the current model of urban development profoundly alters the natural environment, often reducing biodiversity and ultimately threatening human wellbeing. An ecologically based urban planning and design paradigm should consider a more harmonious relationship. Through a systematic literature review of 57 papers, this research identified relevant concepts and theories that could underpin this new paradigm. It revealed a noticeable increase in academic interest in this subject since 2013 and the development of concepts and theories that reflect a more holistic socio-ecological systems approach to urban planning and design based on a transdisciplinary integration and synthesis of research. Seven main themes underpin the academic literature: ecosystem services, socio-ecological systems, resilience, biodiversity, landscape, green infrastructure, as well as integrated and holistic approaches. Six of these can be organised into either a sustainability stream or a spatial stream, representing the foundations of a potential new ecological urban planning and design paradigm that applies sustainability-related concepts in a spatial setting. The final theme, integrated and holistic, includes concepts that reflect the fundamental characteristics of this new paradigm, which can be termed 'urban consonance'.

**Keywords:** urban planning; systematic literature review; ecosystem services; urban consonance

## 1. Introduction

The impact of human activity on the Earth's environmental systems is now so dominant that it is recognised as a new geological age: the Anthropocene, or human-dominated geological epoch [1]. A defining feature of the Anthropocene is urbanisation. In 2015, more than half of the world's population lived in urban areas, and by 2050, it is expected that two-thirds of humanity will call a city home [2,3]. This trend has been even more dramatic in a country like China with a shift of population from rural to urban areas occurring in a relatively short period of time [4].

Cities are responsible for 80% of the greenhouse gas emissions causing climate change. The design of urban areas with increased impermeable surfaces and reduced vegetation also contributes to urban heat island effects, exacerbating heat waves that adversely impact public health [3,5,6]. Cities profoundly alter the natural environment and threaten species diversity and ecosystems through physical changes to land use patterns, fragmentation, and degradation of habitats, the introduction of exotic species and the modulation of natural hydrological, energy flow, and nutrient recycling patterns [4,5,7,8].

Particularly since the 1987 United Nations Brundtland Report's featuring the concept of sustainable development, there has been significant research undertaken in relation to urban sustainability [9]. Despite this, the current model of urban development is unsustainable, threatening human health and wellbeing, and ultimately impacting on the limits of planetary ecosystems [2,10]. The importance of landscape in addressing climate change is often overlooked in urban planning and design and more

often than not landscape elements are considered after the built environment has been constructed [11]. Nevertheless, the role of urban landscapes is considered fundamental to liveable and sustainable cities [12,13]. Landscape is where people and nature interact most acutely, and where ecosystems reside and provide valuable services to people [9]. These ecosystem services include water management, urban cooling, air quality, food production, stormwater and disease control, and recreational, aesthetic, spiritual and psychological benefits [10,14,15]. Green spaces in cities can help to alleviate the effects of climate change, including providing flood protection, shading vegetation for urban cooling, and biomass for carbon storage [16]. For instance, it is estimated that increasing tree canopy cover in Australian cities by 10% could contribute to reducing surface temperatures from paving, walls and roofs by 15% [17].

Proximity to nature and green space can be measured economically in terms of increased property values, tourism revenues, increased air quality, reduced energy consumption and reduced infrastructure costs [18]. For example, the presence of broad-leaved street trees has been found to increase median property prices in Perth (Australia) by almost AUD $17,000 [19]. In Portland (USA) the use of natural elements for stormwater management saved the local government approximately US $60 million [18]. Landscapes can also serve to strongly connect people to place [20]. Cities that are place-oriented are more likely to reduce their ecological footprint, value local ecological features, have strong social capital of networks and trust, and robust urban economies [21].

Concurrent with the developing appreciation of the value of nature in cities is an understanding of an innate human need for contact with nature. Numerous studies have shown the psychological and physiological benefits of proximity to nature and green space such as reducing stress and anxiety, decreasing aggressive behaviour and associated crime levels, faster healing rates for hospital patients, increased physical activity and greater social activity and community bonding [6,16,22].

## 1.1. Application of Ecological Principles in Urban Planning and Design

Urban planning and design are goal-oriented processes that seek to balance social, cultural, environmental, technical and economic considerations within a particular legislative framework [23,24]. The dominant paradigm influencing urban planning and design is modernism [25], which in turn is heavily influenced by scientific rationalism based on a mechanistic, reductionist worldview [26–28]. The consequences of modernism are the planning of cities as separate component parts; the reliance on technology and engineered infrastructure to provide urban functions; the compartmentalization of knowledge; and a dualistic perspective of humans and environment as separate from each other [20,24,25,27,29].

In the 1960s and 1970s, in the context of an increasing focus on environmental issues, scholars and practitioners began to give greater recognition to an ecological approach to urban planning and design [9,24]. The growth of interest in this area has been particularly noticeable in the past thirty years, with a range of theoretical concepts being put forward, including ecosystem services, landscape urbanism, urban ecology, landscape ecology, biophilic design, resilience planning and regenerative design [8,24,30]. A range of tools, frameworks, and assessment systems have also been developed to support the application of ecological principles into building design, landscape architecture and urban planning. An example is the Sustainable Sites Initiative (SITES) for landscape design [4].

Despite these examples of uptake, ecological principles have not yet become mainstream in urban development across the world [24]. A shift is required to bridge the gap between theory and its application in urban planning and design in which landscape sustainability is a key concept [31].

## 1.2. Application of Systems Thinking to Cities

A systems perspective sees the world in a holistic way, looking at the relationships and interactions between parts, predicting their behaviours and seeking to devise integrative solutions that produce desired outcomes [32,33].

There is a growing understanding that cities and urban landscapes are a unique form of human nature integrated system [34]. Viewing cities as socio-ecological systems provides the opportunity for systems thinking to be applied to the planning of cities. For example, [30] notes that systems thinking provides a platform for a more holistic approach in which urban areas, particularly cities, are considered as complex living systems. The challenge of a systems approach is in conceptualising the urban system in a way that does not require complex modelling and can be readily understood by planners and key decision-makers [30].

The purpose of this article is to investigate the key theoretical concepts relevant to the integration of ecological principles with urban planning and design and understand whether they could lead to an emerging ecological paradigm in this area. This research was conducted through a systematic literature review (SLR).

## 2. Methods

The SLR is a scientific approach to identify literature to address specific research questions in a manner intended to minimise bias [35]. The systematic search for, and analysis of, relevant studies are more transparent than traditional narrative literature reviews; and is more likely to result in a broader range of articles that allows for the mapping of specific trends or theoretical directions as well as the ability to identify gaps and areas of uncertainty [35,36]. Bias cannot be entirely eliminated from a SLR as the selection of databases, the application of inclusion/exclusion criteria, the filtering of articles for analysis and the critical appraisal of results all involve a level of subjectivity [35]. However, in a SLR the methodology is explicitly stated, allowing others to assess the author's assumptions, procedures, evidence, and conclusions [36].

While there is no single methodology to carry out a SLR, this research was guided by a number of best practice models, following five distinct steps: problem definition and scope; formulation of the search string; literature search; results and analysis; and discussion and conclusion [37–39].

### 2.1. Problem Definition and Scope

This SLR seeks to identify and map key concepts and theories relevant to the integration of ecological system principles in urban planning and design that could provide the basis for a potential new ecological urban planning and design paradigm.

A taxonomy for literature reviews was adopted to define the search scope, goal, organization, perspective, audience, and coverage [38]. As the objective of this SLR is to understand both the theoretical basis and practical application of ecological principles in urban design and planning, the search included all types of research articles. The goal was to integrate and synthesize the various concepts in the literature to identify the basis for new ecological urban planning and design. The organization of the results was both conceptual and methodological. The intent of the review was to be as objective as possible without favouring a particular perspective. The audience was broad, covering all groups involved in or affected by urban design and planning. An initial scan of available papers revealed the large volume of literature in this field; thus the coverage included only a representative sample of these studies, selected according to the selection criteria described in the next section.

### 2.2. Formulation of Search String

The next step was to identify the more specific search string relating to the research objectives outlined in the introduction.

Potential articles relating to the topics of ecological systems, urban landscapes, and urban planning and development were identified through a preliminary scan of existing databases based on these keywords. The resultant papers were used to establish keywords and associated terms commonly employed in the literature, grouped as shown in Table 1.

**Table 1.** Keywords and associated terms.

| Keywords | Associated Terms |
|---|---|
| Ecology | Ecosystem services, ecosystems, landscape ecology, urban ecology, biodiversity, nature, conservation, wildlife |
| Systems | Systems thinking, systems approach, synthesis, dynamics, thresholds, flows, metabolism, uncertainty, non-linear, circular, holism, integration, transdisciplinarity, resilience |
| Urban | Built environment, residential, green space, landscapes, housing |
| Biodiversity | Biodiversity corridors, wildlife allotments, green corridors, nature corridors, urban wildlife |
| Infrastructure | Green infrastructure, landscape infrastructure, green space, green roofs, green walls, water |
| Landscape | Residential landscapes, urban landscapes, landscape architecture, landscape design, landscape planning |
| Garden | Residential gardens, private gardens, domestic gardens, sustainable gardens, backyards, communal gardens, community gardens |
| Design | Design framework, design tools, landscape design, regenerative design, biophilic design, sustainable design, geodesign |
| Planning | Urban development, sustainable development, urban planning, landscape planning |
| Sustainability | Sustainab*, sustainable development, sustainability assessment, sustainability indicators |
| Social | Socio*, wellbeing, health |

*: Wildcard term to include all the words containing the letters before the *.

A combination of these keywords and string expressions were subsequently tested in several databases, resulting in the following string expression:

((ecolog* OR ecosystems services) AND (urban OR residential) AND (landscape OR garden) AND (systems OR model OR tools OR assessment) AND (planning OR development OR design) AND (sustain OR biophilic OR regenerative OR resilience))

Given the large number of articles resulting from each of the searches, further inclusion and exclusion criteria were developed. The search was limited to peer-reviewed journal articles in electronic databases. Articles were also limited to those in the English language. Books, book sections, theses, reviews and grey literature were excluded from the results.

It is acknowledged that limiting the search to English articles in peer-reviewed journals in electronic databases exposes this SLR to the risk of language and publication bias [35]. This is also relevant to the decision to exclude grey literature from the SLR on the basis of potential lack of research methods stringency. To counter this potential bias, it was decided to include as broad a variety of articles as possible in terms of theories, methods, and city or regional area during the filter process in the literature search phase within the boundaries of the research objectives and problem definition.

*2.3. Literature Search*

Following the development of the search string and its testing in several renowned databases, the following were chosen for the SLR: SCOPUS, ProQuest, ScienceDirect, SpringerLink, and Web of Science.

References were exported into Endnote and filtered for duplicates, resulting in a total of 616 original articles. Titles and abstracts were scanned to identify articles that included the selected keywords. The resulting references, including authors, year of publication, title and abstract, were then exported to an Excel spreadsheet to facilitate filtering and further analysis. The number of articles exported to Excel was 253. In Excel, each article's abstract was reviewed to give priority to those that

were directly relevant to the research objectives. 103 articles were initially identified; however, some of these were grey literature, books, and book sections and were thus excluded. Furthermore, not all articles were available for download and were also eliminated. The resulting final shortlist of articles was 57. A flow diagram (Figure 1) based on the PRISMA 2009 Flow Diagram [40] shows the literature search process of article selection for the SLR.

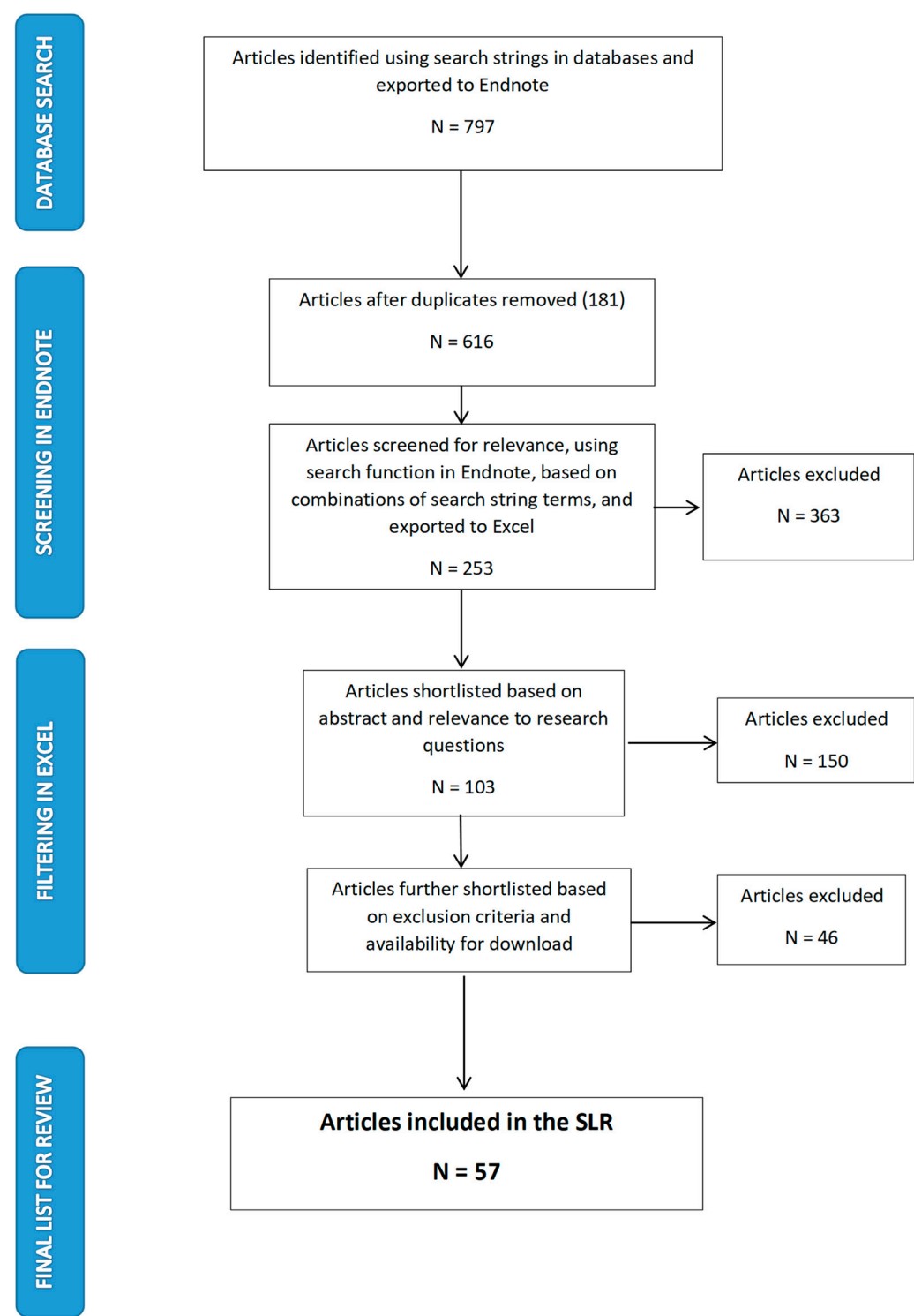

**Figure 1.** Literature search process of selection of articles for inclusion in the SLR (based on the PRISMA flow diagram, [40]).

## 3. Results and Discussion

### 3.1. Reviewed Articles

　　All the articles reviewed were written in the last thirteen years, with the majority (82%) written in the last four years, providing some evidence that the integration of ecological systems thinking with urban design and planning is a relatively new and expanding field of research [24,29]. This is shown in Figure 2.

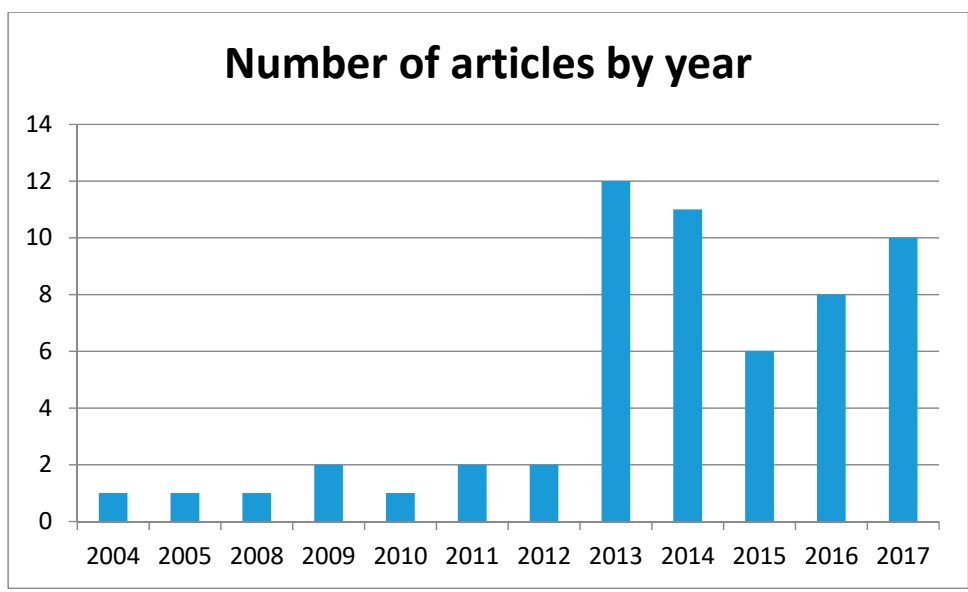

**Figure 2.** Number of peer-reviewed journal articles by year of publication in the SLR.

　　The increasing interest in this field is also reflective of a transdisciplinary synthesis of ideas and concepts from a range of disciplines. The reviewed literature was sourced from 26 journals related to landscape and urban planning, building research, cities, health, environmental science & technology, sustainability, cleaner production, and modelling & software. A full list of the journals and number of articles can be seen in Table 2.

　　The majority of articles (84%) were written by academics located in either Europe (23 articles), the United States (21 articles) or Australia (4 articles). This field is currently dominated by academics located in Western developed countries who bring a particular worldview to their research [14]. The most prominent contribution from non-Western countries was from Chinese academics; 6 of the 9 remaining articles (66%) were by academics based in Chinese universities. The contribution of Chinese academics enables the opportunity to draw on non-Western philosophical traditions that can provide a new perspective to ecological urban planning and design.

**Table 2.** Number of peer-reviewed articles by journal in the SLR.

| Journal Title | Number of Articles |
| --- | --- |
| Landscape & Urban Planning | 15 |
| Landscape Ecology | 5 |
| AMBIO | 3 |
| Ecosystem Services | 3 |

| Journal Title | Number of Articles |
|---|---|
| Land Use Policy | 3 |
| Sustainability | 3 |
| Urban Ecosystems | 3 |
| Ecological Economics | 2 |
| Environmental Science & Policy | 2 |
| Urban Forestry & Urban Greening | 2 |
| Acta Ecological Sinica | 1 |
| Building Research & Information | 1 |
| Cities | 1 |
| Current Landscape & Ecology Records | 1 |
| Ecology & Society | 1 |
| Environmental Modelling & Software | 1 |
| Environmental Research & Public Health | 1 |
| Environmental Science & Technology | 1 |
| Global Environmental Change | 1 |
| International Journal of Sustainable Development & World Ecology | 1 |
| Journal of Environmental Planning & Management | 1 |
| Journal of Cleaner Production | 1 |
| Journal of Environmental Management | 1 |
| Journal of Urban Planning & Development | 1 |
| Landscape Journal | 1 |
| Progress in Physical Geography | 1 |

## 3.2. Identification and Analysis of Key Concepts and Themes

In order to address the main research objective, key concepts and theories that appeared in the reviewed literature were identified and grouped by year of publication and by the number of articles they appeared in. Seven key themes influencing the discussion on ecological urban planning and design were identified, these and their characteristics are shown in Table 3.

**Table 3.** Key themes and their characteristics revealed in the SLR and the number of articles from the total 57 reviewed containing the themes.

| Theme | Characteristics | Articles Containing Theme |
|---|---|---|
| Ecosystem services | Natural Capital<br>Eco-services from nature to humans<br>Provisioning<br>Supporting<br>Regulating<br>Eco-services from humans to nature<br>Conservation<br>Restoration<br>Cultural | 35 |

**Table 3.** *Cont.*

| Theme | Characteristics | Articles Containing Theme |
|---|---|---|
| Socio-ecological systems | Dynamic<br>Integrated human-nature environment<br>Non-linear<br>Complex<br>Non-equilibrium | 17 |
| Resilience | Adaptive capacity<br>Self-organising<br>Virtuous cycle, feedback loops<br>Modularisation | 23 |
| Biodiversity | Variability in species, genetics and ecosystems<br>Ecological connectivity<br>Conservation<br>Habitat | 11 |
| Landscape | Landscape as structure<br>Spatial heterogeneity<br>Multi-scale<br>Landscape connectivity | 19 |
| Green infrastructure | Multi-functional in time and space<br>Multi-object<br>Hybrid of natural and artificial<br>Integrated networks | 23 |
| Integrated and holistic | Ecological wisdom<br>Regenerative<br>Biophilic<br>Permaculture<br>Transdisciplinary | 13 |

### 3.2.1. Ecosystem Services

Ecosystem services (ES) are defined as the benefits the human population derives, directly or indirectly, from biodiversity and ecosystem functions [13,41]. The ES concept was enhanced through the publication of the 2005 Millennium Ecosystem Assessment (MEA) [13]. The conceptual framework for the MEA was centrally focused on the linkages between the world's ecosystem services and human wellbeing, and categorised the benefits of ecosystem functions into four types – provisioning (of products such as food), regulating (of ecosystem processes such as purification of air and water), cultural (of services such as recreation, spiritual and aesthetic) and supporting (of other ecosystem services such as soil formation and nutrient recycling) [9,10,42]. In the reviewed literature, the use of the ES concept and classifications were largely consistent with the MEA framework [9,31,42–45].

Strongly linked with ES is the concept of natural capital. Natural capital (such as soil and living organisms) is the stock of natural ecosystems that yield a flow of ES from nature to human societies [9]. Since the flow of these services is reliant on the function of ecosystems as whole systems, the structure and biodiversity of ecosystems are important components of natural capital [13,31]. A sustainable landscape is one in which the output of ES is maintained, and the capacity of those systems to deliver the same ES for future generations is not undermined [31]. A focus on the anthropogenic benefits of ES and natural capital is useful in that it makes it clear how human needs and wellbeing depend on the environment [4,9]. One way to conceptualise this for application in urban planning is to attempt to geographically map and value the benefits of ES, particularly in economic terms [46,47]. However, this can also be seen as a form of weak sustainability, in that attempts to monetize natural systems may encourage parts of ecosystems to be traded off or discounted according to human perceptions of what has value, or because of the disservices they provide [4,48]. For example, falling leaves from street trees can be perceived as disservices negatively affecting community attitudes [43]. The perception of value and disservices can be highly subjective and variable by different communities and across different environments thus making trade-offs problematic at city or regional scales [14,49].

A variation on the MEA classification of ES types for the urban context is put forward by [50], who distinguishes between types of eco-service from nature to man (provisioning, regulating and supporting) and types of eco-service from man to nature such as the conservation and protection of natural infrastructure in urban areas, the restoration of natural ecosystems to a sustainable state, and cultural services provided by humans to regulate and sustain nature (such as institutional enforcement, spiritual enrichment or eco-tourism). This emphasises a mutual interaction between people and nature that focuses on the creation, restoration, and conservation of urban ecological structures [50,51].

Some authors sought to adapt and broaden the ES concept. For example [42] use the term biophilic services, in order to reveal the benefits and economic value of incorporating natural elements into the built environment. [9] preferred the term landscape services to convey the links between ecosystem services, landscape pattern, aesthetics, values and decision-making.

The ES concept provides a framework through which the ES benefits accrued can be interrogated from multiple perspectives in an approach consistent with systems thinking [52].

### 3.2.2. Socio-Ecological Systems

Socio-ecological systems (SES) theory draws heavily from other theories including systems ecology theory, complex systems theory, resilience theory and sustainability [53]. These have provided important concepts to this field such as complexity, vulnerability and resilience, non-linearity, feedback loops, non-equilibrium dynamics, adaptation and human wellbeing [8,34,53]. SES are an example of a complex adaptive system, which is characterised by many autonomous parts, interacting dynamically in non-linear relationships and at multiple scales with the ability to self-adjust in response to change [9,15,54]. Cities can be considered as a coupled human-nature or linked social-ecological system, highlighting the reliance of human wellbeing on ecosystem services [9,15,24]. While this represents progress from considering social and ecological systems separately, it still permits a compartmentalised approach to occur in terms of substitutability or trade-offs in either ecological or social elements [15,24]. [53] considers this as a mid-point along the way to a fully integrated system of people and nature, described by [15] as a combination of the biophysical exterior and the interior as created and experienced through processes of thought and shared culture. Others liken cities to ecosystems [5,7,55,56]. The application of the ecosystem concept, although drawn from ecology, is used more flexibly in this context by highlighting that the biological component of ecosystems includes people and their social structures and institutional arrangements, buildings and infrastructure, and their interconnectivity [29].

Understanding cities as dynamic, self-organizing systems challenges the modernist planning paradigm. Current urban planning and design thinking promotes an idealised built form based on the idea of equilibrium, or a system predicated on assumptions of stability, efficiency, and predictability [9,34]. However, SES theory requires urban planners and designers to engage with the notion of non-equilibrium, that is a system that is inherently unpredictable and experiencing constant evolution and changes based on multiple non-linear interactions [13,29]. Their activities can therefore only attempt to influence or guide the development of cities in more ecologically desirable directions based on urban ecological knowledge and sustainable principles [13]. For planners and designers, this means there can be no one design or planning approach, or an ideal solution to a given problem [4]. Instead, it requires flexibility and a variety of approaches and solutions, treating every project as uniquely responding to its specific physical, social and economic context [4,34]. It also means viewing designs and plans not as fixed permanent solutions but as opportunities for adaptive learning, building and sharing knowledge in transdisciplinary partnerships across stakeholders [34].

### 3.2.3. Resilience

Increasingly, the concept of resilience is being considered in relation to urban planning and design [4,21]. Two definitions of resilience have been applied in the urban context. The older definition relates to the ability of a system to return to equilibrium after a disturbance [29,53]. This is consistent

with a modernist paradigm in that it focuses on the efficiency and predictability of systems [4]. The second, more recent definition is associated with non-equilibrium complex adaptive systems and emphasises resilience as the ability of a system to absorb and adapt to change while sustaining its fundamental structure and function [29,53]. It is this definition of resilience that is relevant to dynamic, complex and adaptive city systems, and resilience planning has emerged over the past decade as a potential new planning concept [21,29]. Because cities are socio-ecological systems and unpredictable, the sustainability of the system is dependent on its resilience capacity [9,15,20]. For urban planners and designers this means changing the focus from developing an idealised sustainable spatial form of the city to considering how the city can be organised to build its resilience arising from its adaptive capacity [9,34]. The resilience capacity of cities is of particular interest due to the impact of sudden natural disasters such as hurricanes or the long-term consequences of climate change [20,24,34]. [24] note that this requires resilience thinking that facilitates a deep understanding of socio-ecological systems, linking spatial and ecological aspects in a systematic way in order to build resilience capacity. [20] caution this needs to be inclusive of all human and non-human inhabitants of cities, and that focusing exclusively on just the resilience of human society can actually erode resilience in an urban socio-ecological system. Human beings need to understand themselves as part of ecosystems in order to encourage virtuous cycles or feedback loops that produce or enhance ecosystem services and other positive social and ecological outcomes [20].

Strategic, systems-level thinking is needed for planning and design for urban sustainability and resilience in a non-equilibrium context [34]. These can be applied through an iterative, transdisciplinary, adaptive learn-by-doing approach, in which urban plans and projects explore innovative practices and methods, informed by ecological knowledge and research design, that are monitored and analysed for lessons to be applied in future projects [12]. Adaptive urban planning and design are key elements in developing a city's resilience capacity [57].

### 3.2.4. Biodiversity

The concept of biodiversity is a contraction of biological diversity and refers to the variability of, and the complex interactions between, living species, genetic material and ecosystems [58]. The concept became widely used from the 1980s on in response to an increase in interest in biological conservation, and began to be embraced by urban planners and designers in order to improve urban structures as habitats for nature and the protection of ecosystems [24,57,59]. The current approach for considering biodiversity in urban planning is to focus on remnant, biologically dominant patches of habitat such as urban forests and wetlands for rehabilitation and protection [60]. It can be argued however that this ignores the potential for urban biodiversity in other urban spaces such as parks, gardens, road edges and vacant lots [23]. Another consequence of this approach is that biodiversity must compete with the many other priorities of urban planners, such as economic development and transportation [34]. A second approach is reflective of the growing interest in the benefits of ecosystem services. The majority of ecosystem structures and functions, on which ecosystem services depend, are influenced by biodiversity [9,61]. A diversity of species and ecosystems is a key indicator of the health and resilience of urban landscapes and their contribution to quality of life and human health [14,34]. Thus, the ecosystem services concept highlights the importance of biodiversity for human wellbeing. As [31] note this has resulted in a shift in philosophy on nature conservation from being species-centred with an emphasis on site protection approaches to an ecosystem-oriented one focused on an integrated conservation infrastructure.

Landscape connectivity, between habitat patches, wetlands, green space, natural elements, and different ecosystems, is essential for the conservation of biodiversity and ecological flow [5,7,41,62]. Urban planning strategies for landscape connectivity identified in the reviewed literature included green wedges [31,63], green infrastructure, ecological networks [34,44,64], patches, corridors [62,65], domestic gardens [66], vacant or derelict land [60] and green roofs and walls [59].

### 3.2.5. Landscape

Urban landscapes are the scale at which people and nature interact most acutely, and by considering these interactions in a spatially explicit way, it is possible to effectively link local and global sustainability [9]. The landscape approach offers a holistic methodology to define and develop the interface between nature and culture and is thus at the heart of sustainability [31]. Urban studies are therefore increasingly taking a landscape perspective, and arguing the need for landscape and the services it provides to be incorporated into urban planning and design [12,13]. Indeed, landscape is being seen as both a medium and a method for human-nature synthesis in urban design, planning and development [5,60,67]. It provides a common platform for a variety of professions from ecologists to planners to be able to work together in a transdisciplinary way [13].

A number of landscape theories relevant to urban planning and design were noted in the reviewed literature. Three distinct types of theories have been identified by [68], each of which has very different urban planning outcomes:

1.  Design integration theories – these propose that designed landscapes should be integrated into the existing urban context and adapted to the existing urban structure. Phytoremediation-by-design of contaminated sites, and design-with-nature are examples [68].
2.  Ecological integration theories—these propose that natural systems, not designed landscapes should be integrated as support elements within existing urban contexts and processes. An example is New Urbanism theory that originated in the 1980s [4].
3.  Landscape structure theories—these propose that landscape systems, not the built environment, should be the organising principle of urban design and planning [4,68]. Landscape becomes the infrastructure of processes and field of operation [31].

Two further theories were also highlighted as influential in the reviewed literature: urban ecology and landscape ecology. The two theories are closely linked although there are differences in their origins, and thus the perspectives and contributions they bring are slightly different in emphasis. Both bring important concepts to the discussion about ecological urban planning and design. Landscape ecology provides a strong scientific base, concepts, and frameworks for understanding urban biodiversity and the importance of spatial heterogeneity in complex and dynamic urban ecosystems that can be integrated into urban planning and design [9,23,31,34]. Urban ecology seeks to understand the complex and dynamic interactions between socio-economic and natural processes in cities, by considering the whole city as an ecosystem [8,13,55].

Spatial heterogeneity, or landscape pattern, is an important concept in all the landscape related theories discussed in the reviewed literature. It influences biodiversity, ecosystem functions and services, the generation and flow of ecosystem services across a landscape and thus human wellbeing, and is an important component of the adaptive capacity for resilience for the socio-ecological system [4,9,64]. How spatial heterogeneity is conceptualised impacts on the shape and sustainability of urban areas and cities. For example, [60] show how a focus on either ecology in cities or ecology of cities affects the planning and management of urban landscapes. Ecology in cities is a bio-centric approach and thus only focusses on the broad scale spatial elements of urban landscapes that support biological elements, ignoring other potential sources of biodiversity and ecosystems at a smaller scale. In contrast, ecology of cities takes a social-ecological approach to spatial heterogeneity, considering the fine grain of both social and biophysical elements of the urban landscape. This allows for the potential for urban design that encourages more interaction between biotic and abiotic elements as hybrids of vegetation, surface covers, and buildings. The consideration of connectivity between landscape elements is vital as it influences the flows of energy, materials, and biological organisms that underpin the provision of ecosystem services [34,41]. Connectivity of landscapes between multiple geographical scales is also important. Local level landscapes affect and are affected by regional and global sustainability and thus, as noted by [9] landscape sustainability is inherently a multi-scale concept. Multi-scale planning can secure green space against urban development, and enhance connectivity of these spaces from

the city to the regional level [69]. Many ES benefits require this multi-scale approach, for example, to protect and maintain the function of hydrological systems [16]. Multi-scale is also a characteristic of resilient cities, allowing complex and dynamic interactions between socio-economic and ecological processes at different scales to occur that can contribute to the adaptive ability of a city to deal with change and disturbance [7,24]. Applying the multi-scale concept to urban planning and design requires understanding that landscapes function at nested scales and how a design element (e.g., waterway) operates at multiple scales (e.g., from the local site to a regional watershed) [67].

### 3.2.6. Green Infrastructure

The green infrastructure (GI) concept was first put forward in 2006 and has since grown in popularity and even been integrated into urban planning policy, for example, the New York City Green Infrastructure Plan [57,59,70,71]. The GI concept differs from the provision of greenways for aesthetic and recreational purposes by focusing on ecology and on the provision of ecosystem services in cities [12,31]. The description of GI has been broad in the reviewed literature. Some articles described GI as interconnected networks of green spaces that conserve natural ecosystem values and functions that are planned and managed as an integrated system [55,71]. Others included blue spaces such as rivers, wetlands and lakes [56,71]. Still, others take an even a broader perspective, including hybrid and artificial built infrastructure, such as green roofs and walls, and even grey infrastructure such as roads and utilities [12,16,48,64]. Another related term, ecological infrastructure, tended to be preferred by Chinese authors [31,64,72]. Variations in the description of GI can be attributed to the evolution in the concept from a spatial focus on conservation of natural ecosystems for habitat and recreation to a more deliberate technical/functional approach to incorporate ecosystems services infrastructure into the built environment for human welfare, and in support of sustainability [71].

Another key and fundamental characteristic of GI is its multifunctionality that is the provision of multiple ecosystem services that interact within and beyond a shared geographical location [44,55]. Multifunctionality is particularly important in the context of urban planning and design as it allows the provision of multiple ecosystem services in a discrete area of urban land, both spatially and temporally (changing uses in the same space over time or a season) [34]. This potentially can also help to build public support for GI as diverse stakeholders may share an interest in a specific multifunctional landscape [34]. Multifunctional interconnected GI can also support social and economic sustainability in a number of ways, for example by supporting the cultural ES functions offered by landscapes thus valuing humans as part of the ecosystem, encouraging new functions in cities such as food production, and serving as an adaptive strategy for future climatic changes [55]. A multi-object approach to GI planning that includes all forms of public and private blue and green spaces, such as forests, agriculture, playgrounds, golf courses, private gardens, cemeteries, streams, and lakes has also been argued for by Artmann et al. [69].

Despite being promoted for its benefits, the reality of many current examples of GI is that they have been implemented on the basis of a single benefit such as stormwater abatement [44,70]. If poorly designed or managed, GI can also be a source of pollution and compromise urban biodiversity, for example by nutrient runoff from green space into water bodies [64]. Contemporary GI projects such as greenways and restored waterfronts as part of urban redevelopment can also be dismissed as token green interventions in the urban landscape [55]. This is reflective in part of a lack of knowledge and understanding of the dynamic nature of urban ecosystems including their social and ecological interactions, spatially and temporally, and thus a lack of availability of integrated planning models that can evaluate potential multiple synergies and benefits [44,45]. For GI to deliver its ecosystem services it requires continuous links of knowledge and engagement, between people with different duties, responsibilities and aspirations [71]. GI, therefore, provides opportunities for transdisciplinary research, collaboration, and decision-making, provides a nexus between disciplines, brings natural elements into the same framework as other planning concepts, and is applicable at a range of scales [23]. Interest in GI is also driven by the recognition of the role it can play in contributing to the resilience

capacity of cities in the face of climate change [57,73]. The GI concept thus contributes to a more holistic approach to integrating and valuing both biological and human ecology with planning, and therefore supports urban sustainability [55,56].

### 3.2.7. Integrated and Holistic

A prevailing notion of the mechanistic, reductionist worldview underpinning the modernist urban planning paradigm is that humans are separate from, and superior to, nature [15,20,25]. One outcome of this has been a human value system that assumes the right to use ecological resources and change ecological processes for maximum human benefit without limitation [15]. The increasing focus from the latter part of the 20th century on the impacts of human actions on the earth's ecosystems, and the negative consequences for humans, has encouraged a re-examination of the interdependence of humans on the environment [20]. Paradoxically, [20] argue that this has in part resulted in a negative perception of the human relationship with nature, that positions humans like a virus infecting a healthy system, and that nature needs to be protected from humans. This view insidiously contributes to the continuing alienation of humans from their ecological home, and hampers the ability to visualise and actualise a positive transformation towards sustainability [20].

In order to counter this, consideration needs to be given to a more harmonious human–environment relationship that reframes humans as intrinsically part of, and fundamentally dependent on the natural world [15,20,24]. There is evidence that such a new ecological paradigm is emerging, based on a synthesis of older philosophies, and evidence-based findings from new research in ecology, physics, social sciences, sustainability and resilience [13,15,25]. This paradigm is based on a whole systems perspective of socio-ecological systems that emphasizes interconnection, interdependence, adaptability, co-creation and co-evolution, and the reciprocal relationship between humans and nature [15,24].

Biophilic urbanism as a design theory can be traced to the early 2000s and seeks to use natural elements as purposeful design features in the built environment in order to provide the benefits of daily exposure to nature [42,74]. Evidence supports the social, economic and environmental benefits of this exposure such as reduction in stress, increased physical and mental health, greater worker and student productivity and improved urban environments [22,75]. However, as discussed in the reviewed literature, it still tends to be very anthropogenic in focus. For example, [42] use their adapted term biophilic services to explicitly link the benefits of biophilic urban elements for human wellbeing. Biophilic urbanism is also an example of what [68] describes as a design integration theory, or the design of nature into an existing built environment. Permaculture is a design approach more consistent with the systems thinking required of an ecological paradigm. Developed as a concept in the 1970s to foster sustainable agriculture, it has since evolved into a vision of a permanent, sustainable culture based on its core principles of care for earth, care for people, and fair share [15]. Its design principles emphasize the application of locally appropriate methods and solutions based on observation and interaction with natural processes, designing from broad patterns to details, valuing biodiversity, acceptance of feedback and self-regulation, and creative responsiveness to change [15,24]. Another design approach more consistent with an ecological paradigm is regenerative design, a concept introduced by John Lyle in the 1990s [4]. In regenerative design, the landscape becomes the unifying, integrating network of urban form, making use of the self-organization and self-designing capability of natural ecosystems to provide services such as energy, food, water treatment and waste assimilation [15].

A more recent concept called ecological wisdom is emerging out of China [25]. The first International Symposium on Ecological Wisdom for Urban Sustainability was held in China in 2014 with the goal to foster its application in landscape and urban planning [24]. Ecological wisdom distinguishes itself from a modernist approach by advocating for a harmonious human–environment relationship based on associating human values with ecological integrity, and a built environment integrated with nature and natural systems [24]. Three ethical responsibilities for urban planners and designers implementing ecological wisdom have been identified by [24] – to guard the wellbeing of all species and their habitats; to preserve resources for future generations; and to perform competently in

making, selecting and implementing plans and designs as a duty to keep and enhance quality of life. The process of designing and implementing a project is as important as the outcome, with sufficient time required to allow interaction with the community, share knowledge and wisdom, and build ecological integrity as a human value [24].

### 3.3. Historical Analysis

The SLR revealed a number of relevant concepts and theories and identified seven main themes influencing the discussion on ecological urban planning and design. Key characteristics within these themes have been identified and are shown in Table 3. A historical analysis shows broadly how key concepts evolved from those focused on supporting nature in cities through urban planning (conservation of habitat, connectivity and networks), to those focused on the benefits of nature for people (human wellbeing), to more integrated and holistic concepts that consider both nature and people as integral and complementary elements of the urban environment. Figure 3 shows a number of these key concepts plotted along a timeline over the last 50 years. The growth of interest in integrating ecology and urban planning and design is noticeable from the 1980s onward. What is particularly interesting however, is the development of concepts and theories that reflect a more holistic socio-ecological systems approach to urban planning and design from the 2000s onward. This can be explained by the increase in interest and understanding of the negative impact of urbanisation on landscape and global ecosystems, the consequences for human wellbeing, insights from systems theory and the advance of sustainability research and policy development [21]. Figure 3 is strongly suggestive that a potential new ecological paradigm in urban planning is emerging, based on a holistic view of integrated human-nature urban systems.

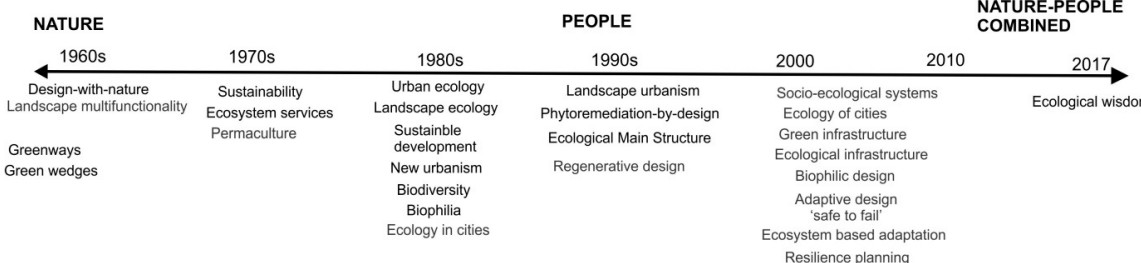

**Figure 3.** Timeline of development of ecological urban planning and design concepts.

### 3.4. Urban Consonance

Six of the themes analysed in the previous section can be further organised into two core streams. The first stream includes ecosystem services, socio-ecological systems and resilience. These three themes are primarily abstract constructs arising within a sustainability theoretical framework. Thus, this can be described as the sustainability stream. The three themes are further characterised by the use and adaptation of their terminology from ecology for example by applying the ecosystem concept to complex, dynamic socio-ecological systems [55]. The second stream includes biodiversity, landscape, and green infrastructure. These three themes are all strongly associated with the spatial element of urban environments and can thus be described as the spatial stream.

These two core streams constitute the foundations of an emergent new paradigm that is the application of sustainability-related concepts in a spatial setting. Sustainability is culturally, socially, environmentally, politically and most importantly spatially context-dependent [9]. Urban planning and design are explicitly spatial disciplines that seek to create urban environments that balance multiple objectives and thus provide the ideal means to integrate and apply sustainability concepts in the urban environment [23,24,56].

Based on the analysis undertaken in the last section, key fundamentals from the six themes consistent with a holistic socio-ecological approach could underpin an emergent new paradigm:

Urban systems are fundamentally complex, dynamic and non-linear integrated systems of people and nature, inherently unpredictable and experiencing constant evolution and changes based on multiple non- linear interactions. Urban planners and designers should attempt to influence or guide the development of cities in more ecologically desirable directions on the basis of sound ecological knowledge and sustainable principles.

Landscape should be the organising principle for urban planning and design, specifically a landscape as structure approach in which urban form is shaped around spatially heterogeneous landscape elements that secure biodiversity and the flow of ecosystem functions and services. Landscape ecology and urban ecology can provide a sound research framework to understand and manage the planning and design of urban landscapes.

Ecosystem services are fundamental for human wellbeing, based on natural capital that yields a flow of services from ecosystems functioning as whole systems. These services flow from nature to humans (provisioning, regulating and supporting) but equally flow from humans to nature (conservation, restoration, and cultural services) in a mutual and interactive relationship.

The health and resilience of ecosystem functions and structures that provide ecosystem services are highly dependent upon the health and diversity of species and ecosystems. Biodiversity requires a high degree of connectivity between natural spaces and different ecosystems for ecological flow, and to conserve habitat.

Multifunctional, multi-scale and multi-object green infrastructure provides the ability to deliver multiple, connected ecosystem services into the built environment both spatially and temporally. It can include integrated networks of green and blue spaces, as well as hybrid structures of artificial and natural elements such as green walls.

Cities are socio-ecological systems and unpredictable, hence the sustainability of the system is dependent on its resilience capacity. This needs to be inclusive of all human and non-human inhabitants of cities in order to encourage virtuous cycles or feedback loops that produce or enhance ecosystem services and other positive social and ecological outcomes. Thus biodiversity and ecological connectivity are important to resilience capacity.

The final theme, Integrated and Holistic includes concepts that incorporate many of these fundamentals and thus can be seen as consistent with this emergent new paradigm. It includes ecological wisdom, permaculture and regenerative design. Biophilia is also included because it is consistent with a more holistic and integrated approach although some inconsistency was noted in the reviewed literature in how it has been applied to urban design and planning. These concepts are interdisciplinary in their approach, looking to ideas from new research in a range of ecological and social sciences, older philosophical holistic notions of harmony and a reciprocal relationship between humans and nature, as well as the wisdom of local experience and knowledge. They also emphasise the application of locally appropriate place-based methods and solutions.

Figure 4 illustrates how the themes influence this potential new ecological urban planning and design paradigm. This new paradigm can be termed urban consonance and reflects a harmony or agreement between nature and people, and describes the harmony of the evolution of key ecological urban planning and design through interdisciplinarity and ongoing synthesis.

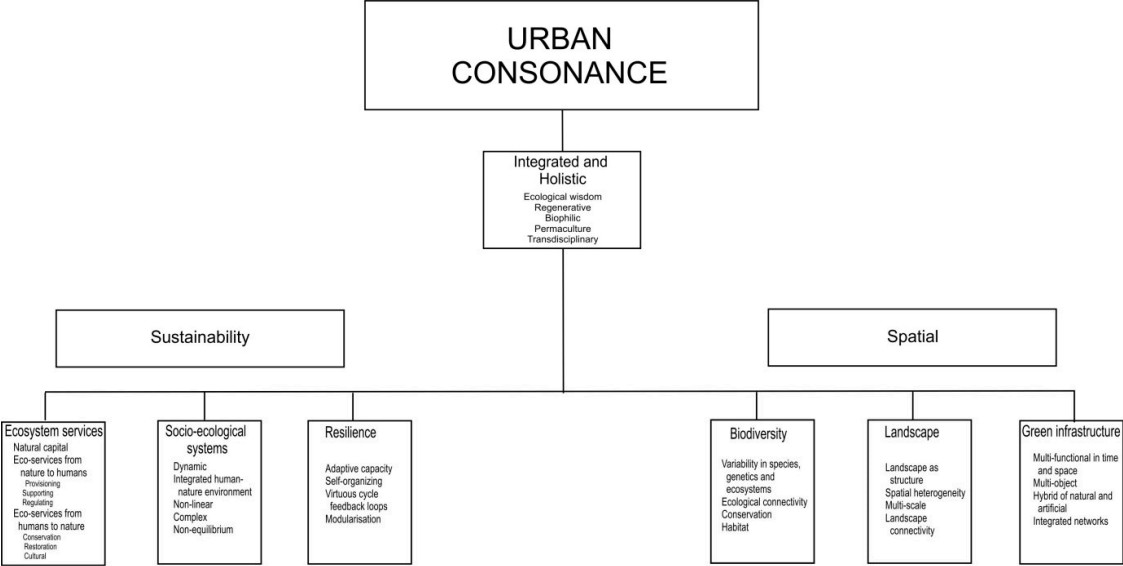

**Figure 4.** Themes and characteristics in Urban Consonance.

## 4. Conclusions

Urbanisation is a defining feature of the modern human-dominated geological age. However, the prevailing model of urban development profoundly alters the natural environment, reduces biodiversity and threatens human wellbeing. Despite a growth in interest in applying an ecological approach to urban planning and design, particularly over the past thirty years, this has not become mainstream in practice and the negative impacts of urbanisation continue. It has been argued that this is due to a modernist urban planning paradigm that sees humans as separate from, and superior to, nature. This has resulted in a human value system that assumes the right to use ecological resources and change ecological processes for human benefit without limitation as well as a reliance on technology and engineered infrastructure to provide urban functions and the compartmentalisation of knowledge. A new urban planning and design paradigm is needed based on a more harmonious human–environment relationship, acknowledging the importance of landscape, and understanding cities as complex, dynamic socio-ecological systems.

Using a systematic literature review, this article identified seven key concepts and theories in a representative sample of the academic literature that could form the basis of an emergent new ecological urban planning and design paradigm. These concepts were arranged under either a sustainability theme or a spatial theme, thus identifying the foundations for an urban planning and design paradigm that applies sustainability-related concepts in a spatial setting. Fundamental characteristics and principles consistent with a holistic, socio-ecological approach that emphasises multifunctional landscapes as the organising principle for urban planning and design, and the role of biodiversity and ecosystem services for human wellbeing and the resilience capacity of cities were also identified. These key characteristics and principles can be seen as the elements of a potential new emergent ecological urban planning and design paradigm called urban consonance.

It is acknowledged by the authors that the sample size and search selection criteria may have limited the literature reviewed. However, to counter this a broad range of articles have been analysed to cover the theories, methods and regional areas. The initial scan of the literature revealed a large volume of potential literature to be reviewed, potentially in the thousands. It was therefore decided to limit the search to peer-reviewed journal articles in electronic databases only. This is intended to provide the rigorous peer-reviewed theoretical and evidence base for a review of ecological principles in urban planning and design. Future reviews could include policy or technical documents used by governments in the field as well as focusing on other themes and conceptualisations in urban

development such as inequalities in social systems, the role of multiple agents in the urban planning process, the role of residents in policy planning and environmental justice issues. Further exploration of the Urban Consonance concept and a thorough exploration of its incorporation into urban planning is also recommended.

Incorporating connectivity for urban biodiversity and ecosystem functions into the planning of urban spatial form requires a better understanding of the functions and services of biodiversity for human wellbeing [34,56]. While there is growing research into urban ecosystems such as long term projects in Baltimore and Phoenix, there is the need for research into the specific links between biodiversity and the delivery of ecosystem services in urban areas [4,14]. In addition, there needs to be better transdisciplinary links between research scientists and urban planning and design and other professionals in order to ensure biodiversity protection is more widely accepted and prioritised in urban planning and design [12,34].

**Author Contributions:** Conceptualization A.H., G.M.M. and J.J.B.; methodology A.H.; formal analysis A.H.; investigation A.H.; data curation A.H.; writing—original draft preparation A.H.; writing—review and editing, A.H., J.B., G.M.M., J.J.B. and C.E.; visualization A.H. and J.B.; supervision G.M.M. and J.J.B.; project administration A.H., G.M.M. and J.J.B.; funding acquisition G.M.M. and J.J.B.

**Funding:** This research is funded by the CRC for Low Carbon Living Ltd. (Project number NP2006) supported by the Cooperative Research Centre's program, an Australian Government initiative.

**Conflicts of Interest:** The authors declare no conflict of interest.

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
