# Peer review of "Ecological Urban Planning and Design: A Systematic Literature Review"

_sustainability, doi:10.3390/su11133723_

Round 1
Reviewer 1 Report
The article reveals methodological rigor in bibliographic research and in the interpretative synthesis regarding 7 topics that are identified as central to the international debate.
The approach is certainly noteworthy for the main purpose of this article: to investigate the key theoretical concepts relevant to the integration of ecological principles with urban planning and design.
For the secondary objective regarding the application to "residential urban developments" or "urban spaces", I think it could be better explored in another article, also referring to some best practices. In this second case, in particular, the different approaches at the international level necessarily require a deepening that the space of the article does not allow and that risks making this second theme appear inadequately treated, if it is compared with the first research topic.
Reviewer 2 Report
Although it presents a very interesting topic, the article has some shortcomings which require methodological improvements or clarification on part of the authors.
The abstract could be a bit more informative by giving some exact information (e.g. data or numbers) on the methodological aspects which lead to the conclusion (no. of papers, e.g.).
Ln 32 – double use of the word dramatic
Ln 34-35 – one could argue that researches on urban sustainability are to be found also before the Brundtland report
The first part of the introduction section is pretty chaotic, using a series of concepts associated with urban environments – landscape, green spaces, nature in cities…
1.1. application of ecological principles.. it is not clear where these principles should be applied in the urban planning and design… authors talk about a range of theoretical concepts and a range of tools, even offering some examples, but why have they not become mainstream? Or is the situation general? In certain parts of the world I would not dare to say mainstream, but they are considered at a higher level than in others.
Ln 102 -106 could be strengthened a bit by better differentiating between the purpose of the article and the objectives of the article
I did not understand exactly how did step 2.2 of the methodological sections worked out? The authors state they identified articles through a preliminary scan of existing databases and then used them to establish keywords and associated terms. How many articles did they selected? Did they use their keywords? Needs clarification.
If the authors excluded from the analysis for being irrelevant more articles than they kept in both phases (363>253, 150>103) isn’t this a major limitation of the methodological process? Is the samples size of 57 articles finally analysed relevant enough to draw conclusions?
Of course Figure 2 shows the increase in the number of articles per year, but I would dare say could be applied to pretty much all modern scientific concepts, simple cause the absolute number of articles published increased from 2004 to 2017
We have no information on the occurrence and distribution of themes and characteristics from table 3 among analysed articles. How many of them presented green infrastructure per example?
The description of the themes should not put that much emphasis on the description of the terms (as they are all very well-known ones) but more on the connection with urban planning and design authors found in the papers.
How did the authors realized the historical analysis if they only had articles starting with 2004 (as it is presented in the figure with the distribution of articles per year). If they used other sources than the database of articles they should explain that a bit. The section of historical analysis is a plus of the paper and I appreciate it, just not sure on how it was made.
The authors should strengthen a bit in the final of the paper the limitations of the study and potential for future research.
Also, maybe could be discussed a bit more that the analysis is based on papers, so mainly academics view on the concepts, but urban planning and design uses a lot of policy documents and technical ones developed by practitioners of the fields. Could we related the groups and their use of concepts or methods?
Reviewer 3 Report
“A brief summary (one short paragraph) outlining the aim of the paper and its main contributions.”
The aim of this article is “to investigate the key theoretical concepts relevant to the integration of ecological principles with urban planning and design and understand whether they could lead to an emerging ecological paradigm in this area”. A secondary aim is to discuss the application of ecological principles to residential urban developments.
The method used is a “systematic literature review” (SLR). The authors argue for the need of a new ecological paradigm in urban planning.
On the basis of the systematic literature review, highlighting seven “key themes” influencing “ecological urban planning”, the authors sees the emergence of a new ecological urban an planning and design paradigm. This paradigm is labelled “urban consonance”.
“Broad comments highlighting areas of strength and weakness. These comments should be specific enough for authors to be able to respond.”
The topic of this article is timely and highly relevant. There is certainly a strong need for more holistic and transdisciplinary approaches in urban planning and design, which recognises the complexities and dependencies in human-nature-human relationships. The article reviews and gives and interesting overview of seven important themes which can broadly be associated with ecological principles, which in turn potentially can inform urban planning and design. However, there are some major weaknesses that make me suggest rejecting the article in its current form. These weaknesses are related to 1) Novelty/Contribution and 2) Definitions and theorization of key concepts.
1) Novelty/Contribution
The novelty/contribution of the paper is not clearly shown or argued for. One main finding is that the reviewed articles, with focus on urban ecology perspectives, are fairly recently published and that there are an increasing number of publications over the years, growing from 1-2 articles per year (2004-2012) to 6-12 articles per year (2013-2017). This finding contributes to the conclusion that there is an emerging ecological urban planning paradigm. However, in order to get a better picture of the role and relative weight of this new eco-urban paradigm, I would have preferred a more in-depth discussion about the meaning of these numbers.
Are these numbers exceptional (there is a general growth in peer-review article publications!)?
What kind of growth patterns would we find in other, and perhaps “competing” paradigms, related to urban planning and design: “innovation”, “mobility”, “transit oriented development”, “densification”, “urban renewal”, “smart growth” etc.? Can you say something about the themes in the articles that was taken out of the review?
Lastly, what is the relationship and exchange between the academic interest in concepts such as biodiversity, ecosystem services, socio-ecological systems and “on the ground” planning and design? While many of these “key themes” are new – the article seems to understate the fact that parks, urban forests, tree-lined streets and channels, gardening have been important (planning) features in cities for centuries. Is it really the lack of (synthetized) knowledge in the field of ecological urban planning that is missing, or are their other obstacles and prioritisations that makes the current urban model unstainable?
2) Definitions and theorization of key concepts
Although there is an interesting review of seven key themes found in the literature review, the most important concept in the article is poorly defined and theorised: namely “urban/urbanization” (“urban system”, “cities”, “urban landscape” and “urban areas”). We get to know that a majority of people in the world are living in urban areas. We also get to know that modernism is the dominant paradigm in urban planning and that cities, from a “socio-ecological system perspective”, ought to be seen as “self-organizing” and “dynamic systems”.
Other than that there little or no reference to important discussions within urban development today: increasing inequalities and residential segregation, environmental justice perspectives concerning among other thing the inequitable distribution of green space, growth of neo-liberal planning and privatization of public spaces. Although these perspectives put more weight on the “social” side of the sustainable trialectic, they arguable need to be taken in the analysis as these perspective gives a more full-fledged picture of how the “the current model of urban development is unsustainable” (line 36).
The present theorization of the “urban” seems, without these broader, critical reflections, a bit naïve and reductionist. No people, actors, organizations or power-relations are highlighted, except a small reference to participatory approaches in the end of the article (line 614-618). Further, there are no clear differences highlighted within or between cities in different regions and countries. E.g. on line 38 it says that cities are responsible for 80 % of greenhouse gas emission: can “cities” really be seen as responsible agents, and in that case, what is the specific role of urban planning and design in contributing to/reducing these emissions? Instead, cities are first and foremost describes as “self-organizing system” and the main actor that is visible in the article are “urban planners and designers”, in turn hiding a broad spectrum of professionals: engineers, landscape architects, architects, economists, geographers.
“System” is another concept or theoretical perspective that needs further conceptualisation. On line 92-93, a system perspective is described as “looking at relationships and interaction between parts, predicting their behaviours” (line 93). This statement sounds like a very modernistic (“social engineering”) way of describing human actors, something that is contradictory given the critique put forward in the article of the modernistic paradigm in urban planning. The aim in the article of fostering more “harmonious” human- environment relations also seem to clash with the ideas promoted in socio-ecological system thinking about ecology as something “unpredictable” and in “constant evolution and changes based on multiple non-linear interactions”. Further, the system perspective is portrayed as a holistic and transdisciplinary perspective, a claim that can be debated (see e.g. Olsson et al. 2015. Why resilience is unappealing to social science: Theoretical and empirical investigations of the scientific use of resilience). In order to reach “urban consonance” (also poorly described), which of course is a beautiful intention; the article needs to better recognize, and explicitly try to find bridges to overcome, the different ontological and political positions that clashes when different key themes/concepts, theories and world-pictures meet and are negotiated by actors with different power positions.
Round 2
Reviewer 2 Report
Thank you for considering and integrating our comments.